



# Multidecadal variability in seasonal mean sea level along the North Sea coast

Thomas Frederikse[1] and Theo Gerkema[2]

[1]Jet Propulsion Laboratory, California Institute of Technology, 4800 Oak Grove Drive, Pasadena, California, USA
[2]NIOZ Netherlands Institute for Sea Research, Department for Estuarine and Delta Systems (EDS), and Utrecht University, PO Box 140, 4400 AC Yerseke, The Netherlands

**Correspondence:** Thomas Frederikse (thomas.frederikse@jpl.nasa.gov)

**Abstract.** Seasonal deviations from annual-mean sea level in the North Sea region show a large low-frequency component with substantial variability at decadal and multi-decadal time scales. In this study, we quantify low-frequency seasonal variations from annual-mean sea level and look for drivers of this variability. The amplitude, as well as the temporal evolution of this multi-decadal variability shows substantial variations over the North Sea region, and this spatial pattern is similar to the well-
known pattern of the influence of winds and pressure changes on sea level on higher frequencies. The largest low-frequency signals are found in the German Bight and along the Norwegian coast. We find that the variability is much stronger in winter and autumn than in other seasons, and that this winter and autumn variability is predominantly driven by wind and sea-level pressure anomalies which have their cause in large-scale atmospheric patterns. For the spring and summer seasons, only a small fraction of the observed variability can be explained by local and large-scale atmospheric changes.

Large-scale atmospheric patterns have been derived from a principal component analysis of sea-level pressure. The first principal component of sea-level pressure over the North Atlantic Ocean, which is linked to the North Atlantic Oscillation (NAO), explains the largest fraction of winter-mean variability for most stations, while for some stations, the variability consists of a combination of multiple principal components.

The low-frequency variability in season-mean sea level can manifest itself as trends in short records of seasonal sea level.
For multiple stations around the North Sea, running-mean 40-year trends for autumn and winter sea level often exceed the long-term trends in annual mean sea level, while for spring and summer, the seasonal trends have a similar order of magnitude as the annual-mean trends. Removing the variability explained by atmospheric variability vastly reduces the seasonal trends, especially in winter and autumn.

## 1 Introduction

Analyses of sea-level records, with a view to deducing trends and their causes, as well as sea-level projections commonly focus on annual-mean values (Wahl et al., 2013; Piecuch et al., 2016; Slangen et al., 2017, e.g.). However, next to interannual sea-level variability, which is captured by annual-mean sea level, season-mean sea level (e.g. winter-mean or summer-mean) could have its own variability on top of the annual-mean variability. In this study, we quantify this seasonal sea-level variability in





the North Sea region, and look into its causes. It has already been demonstrated that for the Southwestern North Sea, different seasons show distinct variability patterns: Dangendorf et al. (2013) demonstrated the difference between annual-mean and seasonal variability from the Cuxhaven tide-gauge record. In particular, variability in spring and summer (which were broadly similar) diverged strongly from the autumn and winter seasons. Hence, variability of annual-mean sea level is not necessarily

representative for variability of seasonal sea level. In the case of Cuxhaven, the disparities were almost entirely explained by local atmospheric forcing (wind stress and atmospheric pressure).

While the common variability in and around the North Sea on decadal and longterm time scales is mostly driven by the baroclinic response to remote longshore wind stress (Dangendorf et al., 2014a; Frederikse et al., 2016a), local atmospheric forcing is known to cause large and and localized interannual sea-level signals in the North Sea (Marcos and Tsimplis, 2007;

Dangendorf et al., 2014a; Frederikse et al., 2016b). Therefore, regional variations in seasonal sea-level variability are to be expected. Zonal winds have been shown to affect sea level much more along the Dutch and German coasts than along the British coast, where atmospheric pressure is relatively more important (Dangendorf et al., 2014a; Frederikse et al., 2016b), although the sea-level response to atmospheric pressure often deviates from the inverse barometer effect (Woodworth, 2017b). For the German and Dutch coast, it is well known that a large part of the observed sea-level variability in the winter months can

be explained by the North Atlantic Oscillation (NAO) (Wakelin et al., 2003; Yan et al., 2004), which mostly acts through wind forcing (Chen et al., 2014). The NAO contains a strong multi-decadal component, which results in multidecadal winter-mean sea-level variability in this region (Dangendorf et al., 2012). However, the North Atlantic Ocean does not explain all winter-mean atmospheric variability, and new atmospheric proxies have been proposed: Dangendorf et al. (2014b) uses a proxy based on shifted centers of action, while Chafik et al. (2017) uses the combination of teleconnection patterns, including the NAO,

the East Atlantic Pattern (EAP) and Scandinavia Pattern (SCAN) to explain a larger factor of the local wind and pressure forcing along the east coast of the North Atlantic Ocean. These indices characterize the prevailing large-scale patterns in the atmosphere, but on a regional level, they translate into the actual atmospheric agents that induce local sea level variations: wind stress and atmospheric pressure.

The difference between seasonal and annual mean sea level difference has a strong multi-decadal component, which mani-

fests itself as trends in records that span multiple decades (Marcos and Tsimplis, 2007). For several stations along the Dutch coast, Gerkema and Duran-Matute (2017) also suggested that this variability causes differences in trends estimated over 100-year long tide-gauge records.

The purpose of this paper is twofold: first, we want to quantify the multi-decadal variability in season-mean sea level for the North Sea region, and we want to investigate which fraction of the seasonal variability is caused by local and large-scale

atmospheric forcing. As an explanatory factor for local forcing, we will look into winds and atmospheric pressure. Those local forcing agents do contain a signal that is linked to large-scale atmospheric oscillation patterns. We will investigate whether changes in these large-scale atmospheric patterns are responsible for the multi-decadal variability in seasonal sea level.



## 2 Data and methods

For this study, we use monthly-mean sea-level observations from 33 tide-gauge stations around the North Sea, the Norwegian coast, and the English Channel. The data has been obtained from Permanent Service for Mean Sea Level database (PSMSL, Holgate et al., 2013). We only use stations that are not flagged for possible problems and for which that data is provided in a Revised Local Reference (RLR) to avoid stations with unstable datums. For Trondheim and Aberdeen, two individual tide-gauge records have been merged into a single records by adjusting both records to the mean over period where both records overlap. Figure 1 shows a map with all the tide-gauge stations used in this study and the periods over which the stations have data. We limit our tide-gauge data to the period 1890-2014 to avoid the inclusion of the sea-level jump that is apparent in many Dutch tide-gauge stations around 1885. From that year, monthly mean sea level is based on mean sea level readings rather than mean tide level readings, which could result in a jump in the monthly data (Woodworth, 2017a). According to the PSMSL documentation, a correction has been applied to avoid this jump, but the jump is not apparent in some neighbouring stations, and as such, suspect. We only consider years for which at least 10 months of data is available. Starting from the monthly tide-gauge data, we compute seasonal sea-level anomalies as follows: first, we remove the annual-mean sea level from the monthly data. To ensure that each season consists of consecutive months, each year runs from December the year before until November. Then, gaps of two months and shorter are linearly interpolated. The resulting monthly time series is then separated into seasonal deviations for four seasons: winter (December, January, February, DJF), spring (March, April, May, MAM), summer (June, July, August, JJA), and autumn (September, October, November). The monthly sea-level data is averaged over each season, which results in four sea-level anomalies per year, one for each season. The resulting sea-level time series are thus seasonal deviations from annual mean sea level. Annual sea level can be affected by a large deviation in a particular season (e.g. a winter with very high sea level results in a higher annual mean), which in that case will result in an anomaly with an opposite sign during the other seasons. We have followed this approach instead of removing the linear trend or the low-frequency component from the tide-gauge data because in this region, the aforementioned baroclinic response to longshore wind forcing causes a large interannual variability signal, which would leak into the seasonal anomalies if only the linear trend or low-frequency variability instead of the annual mean is removed.

To obtain continuous records of wind stress and sea-level pressure anomalies, we use monthly-mean output from the NOAA twentieth-century reanalysis project version V2C (Compo et al., 2011) and average to get seasonal anomalies. We compute wind stress from the 10-meter wind speeds using the following relation

$$\tau_u = \rho_{\text{air}} C_D u \sqrt{u^2 + v^2} \tag{1}$$
$$\tau_v = \rho_{\text{air}} C_D v \sqrt{u^2 + v^2}, \tag{2}$$

where $\rho_{\text{air}}$ the density of air, $u$ and $v$ the zonal and meridional 10-meter wind velocity, and $C_D$ the drag coefficient, which is parametrised following Pugh and Woodworth (2014):

$$C_D = 0.8 + 0.065 \sqrt{u^2 + v^2} \tag{3}$$



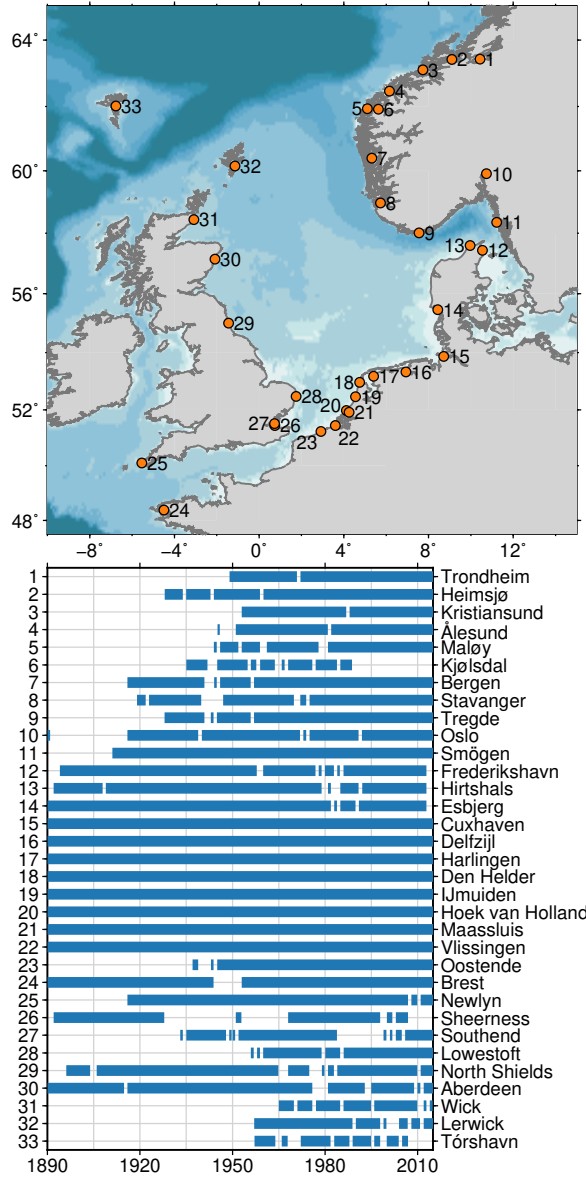

**Figure 1.** Locations of the tide-gauge stations and the availability of data at each station over the period 1890-2014. The numbers in the upper panel correspond to the numbers in the lower panel.





We parametrise the effects of seasonal local wind stress and sea-level pressure on sea level as a linear model:

$$\eta_{\text{obs}} = \alpha_0 + \alpha_1 p'(t) + \alpha_2 \tau_u(t) + \alpha_3 \tau_v(t) + \epsilon(t). \tag{4}$$

In this model, $\eta_{\text{obs}}$ is the observed seasonal sea-level deviation, $t$ is the time of the observation, $p'(t)$ is the local sea-level pressure anomaly, $\tau_u(t)$ the zonal wind stress, $\tau_v(t)$ the meridional wind stress, and $\epsilon(t)$ the residual. We obtained the pressure anomalies and wind stress values by taking the model value with the highest correlation coefficient within a 500km radius around each tide-gauge station. We solve this system using ordinary least squares, which gives us the regression parameters $[\alpha_0 \ldots \alpha_3]$. We test for the significance of each regressor using a $t$-test statistic, and only include regressors, when the accompanying 95% confidence interval does not cross zero.

To obtain a link between coastal sea level and large-scale atmospheric patterns over the North Atlantic Ocean, we have computed the three leading principal components of the sea-level pressure field from the 20th Century Reanalysis following the procedure described in Chafik et al. (2017). The gridded sea-level pressure field between $[80W - 50E,\ 30N - 80N]$ is selected, the seasonal cycle is removed, and from the resulting field, the three leading empirical orthogonal functions and associated principal components have been computed. We have chosen the method from Chafik et al. (2017) over selecting the original indices to obtain a coherent set of large-scale atmospheric variability from a single data source.

As with the effects of local winds and pressure changes, we compute the effect of large-scale atmospheric variability on local sea level using a linear regression model

$$\eta_{\text{obs}} = \beta_0 + \beta_1 \text{PC}_1(t) + \beta_2 \text{PC}_2(t) + \beta_3 \text{PC}_3(t) + \epsilon(t), \tag{5}$$

in which $\text{PC}_n(t)$ is the $n$th principal component. The regression coefficients, $[\beta_0 \ldots \beta_3]$ are estimated using ordinary least squares.

To assess the performance of the model, we use the fraction of explained variance $R^2$, which is defined as:

$$R^2 = 1 - \frac{\text{var}(\eta_{\text{obs}} - \eta_{\text{exp}})}{\text{var}(\eta_{\text{obs}})} \tag{6}$$

with var() the variance operator and $\eta_{\text{exp}}$ the sea-level deviations explained by the regression models. To obtain information on low-frequency variability, we use a second-order Butterworth low-pass filter with a cutoff period of 10 years. Note that the regression models are applied to unfiltered data, and that the filters have been applied as a post-processing step.

## 3   Results

The first objective of this paper is to quantify low-frequency seasonal variability for each season, which is shown in Figure 2. This figure depicts the standard deviation of the low-pass filtered seasonal sea-level time series, which is a measure of the typical amplitude of the multi-decadal variability in seasonal sea level. The figure shows that the low-frequency seasonal variability in winter and autumn is generally larger than in summer and spring for most stations, while the winter-mean variability being the largest. The amplitude of low-frequency winter and autumn variability shows a clear regional pattern: high variability can





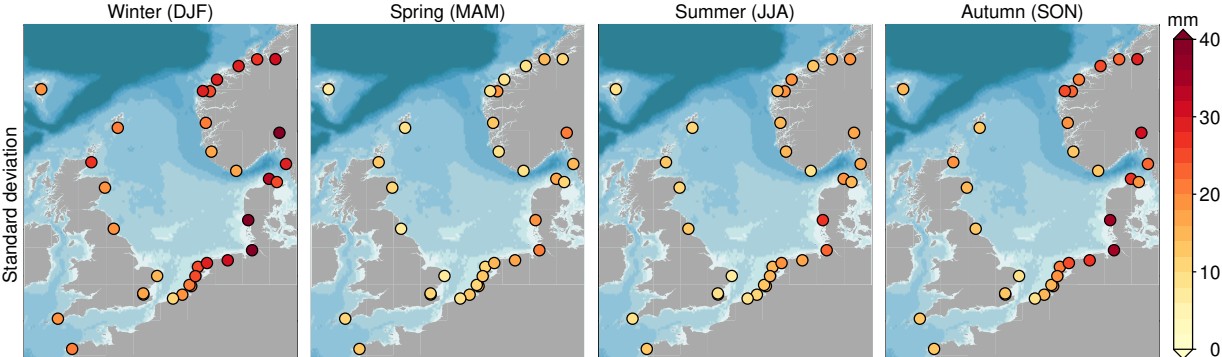

**Figure 2.** Standard deviation of the seasonal sea-level anomalies for each season after applying a 10-year low pass filter at all tide-gauge locations.

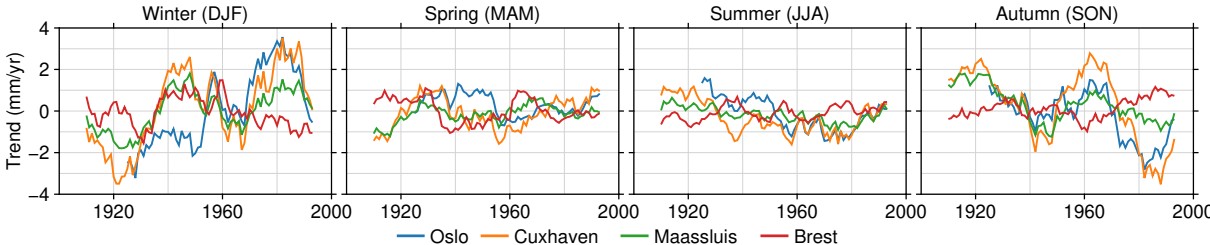

**Figure 3.** Running-mean trends in seasonal sea level deviations for four stations, using a 40-year window. Trends are only shown for time windows with at least 30 years of data. Note that these trends have been computed from the time series without the low-pass filter applied.

be found in the German Bight, the Skagerrak between Norway and Denmark, as well as along the Norwegian coast towards the North, while for the southern North Sea, Brest, Newlyn, and the British coast, this variability is smaller. Although the low-frequency variability in spring and summer is substantially smaller than in winter, the spatial patterns for each seasons show some similarities: also in spring and summer, the variability is highest for the stations surrounding the German Bight. The

5   Southeastern North Sea is a hotspot for low-frequency variability in the region, but the seasonal differences between sea-level variability Dangendorf et al. (2013) found for the Cuxhaven station are not a purely local phenomenon. This low-frequency variability can be interpreted as trends when short records are used. To quantify typical trends that could emerge from low-frequency variability in season-mean sea level, we have computed 40-yr running mean trends in seasonal sea level for four stations. The resulting trends are depicted in Figure 3. For all seasons, the trends in seasonal sea level can reach values in

10   the same range of the long-term trend in mean sea level, which is typically in the order of 1-2 mm/yr for this region (Wahl et al., 2013) with the largest 40-year trends occurring during autumn and winter. For Oslo and Cuxhaven, the seasonal trends sometimes exceed 4 mm/yr, which is about twice the rate of the secular trend.





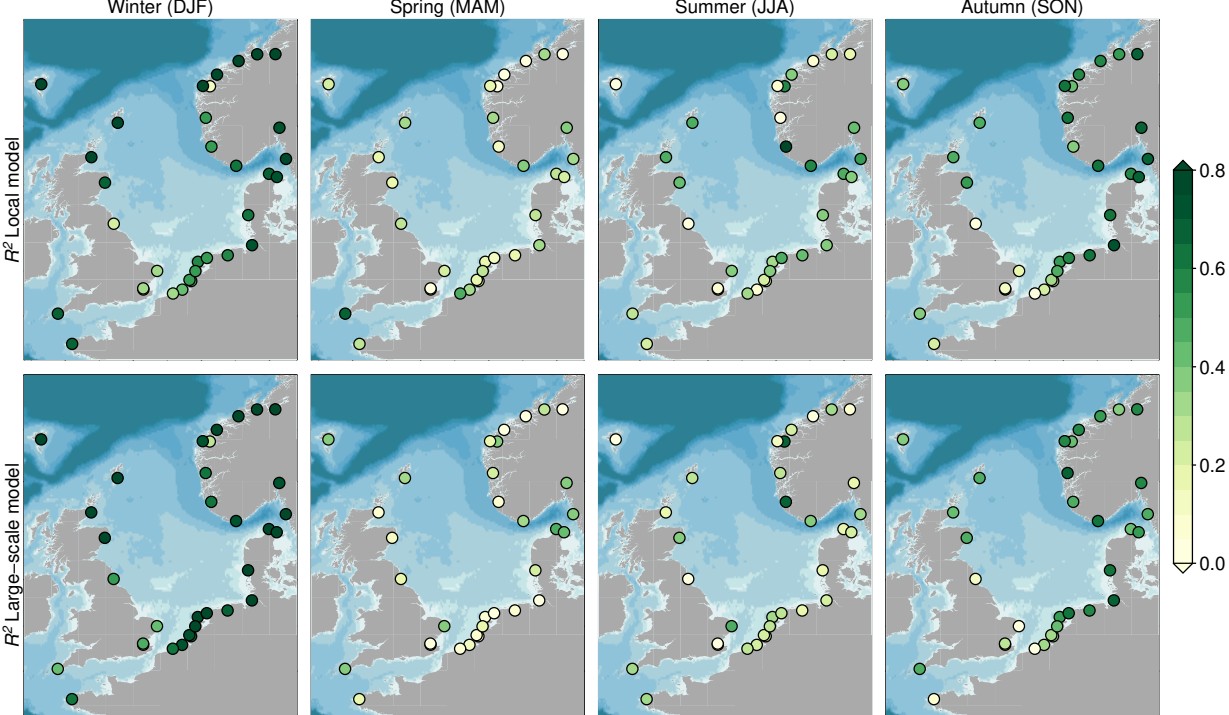

**Figure 4.** Fraction of 10-year low-pass filtered seasonal variance ($R^2$) explained by the local (top) and large-scale model (bottom) for each station.

To determine whether local wind and pressure changes are responsible for this variability, we compute the fraction of explained variance ($R^2$) of the local regression model (Equation 4) after applying a 10-year lowpass filter to both the seasonal sea-level deviations and the solution to the regression model. The results are depicted in the top row of Figure 4.

In autumn and winter, when the low-frequency variability is highest, the local regression model explains a large fraction
of the variability for most stations. Especially in the winter (DJF) season, the model explains the majority of the observed variability ($R^2 > 0.5$) for most stations. In spring and summer, generally only explain a small part of the variability can be explained, which suggests that the long-term seasonal variability, which is already much smaller than the winter and autumn variability, is not predominantly driven by wind- and pressure changes.

Since the local wind and pressure variability is known to be influenced by large-scale atmospheric pattern such as the NAO
and EAP, the next step is to investigate whether the multi-decadal variability in seasonal sea level is also driven by these large-scale atmospheric variability patterns. To this end, we use the three leading principal components of surface pressure variability, as described in section 2. These principal components and their associated empirical orthogonal functions represent the major patterns of atmospheric variability and are displayed in Figure 5. As such, they share characteristics with well-known atmospheric teleconnection patterns. The distinct north-south pattern of the first principal component resembles the
North Atlantic Oscillation (Hurrell et al., 2003), while the second and third PC are akin to the East Atlantic Pattern and the





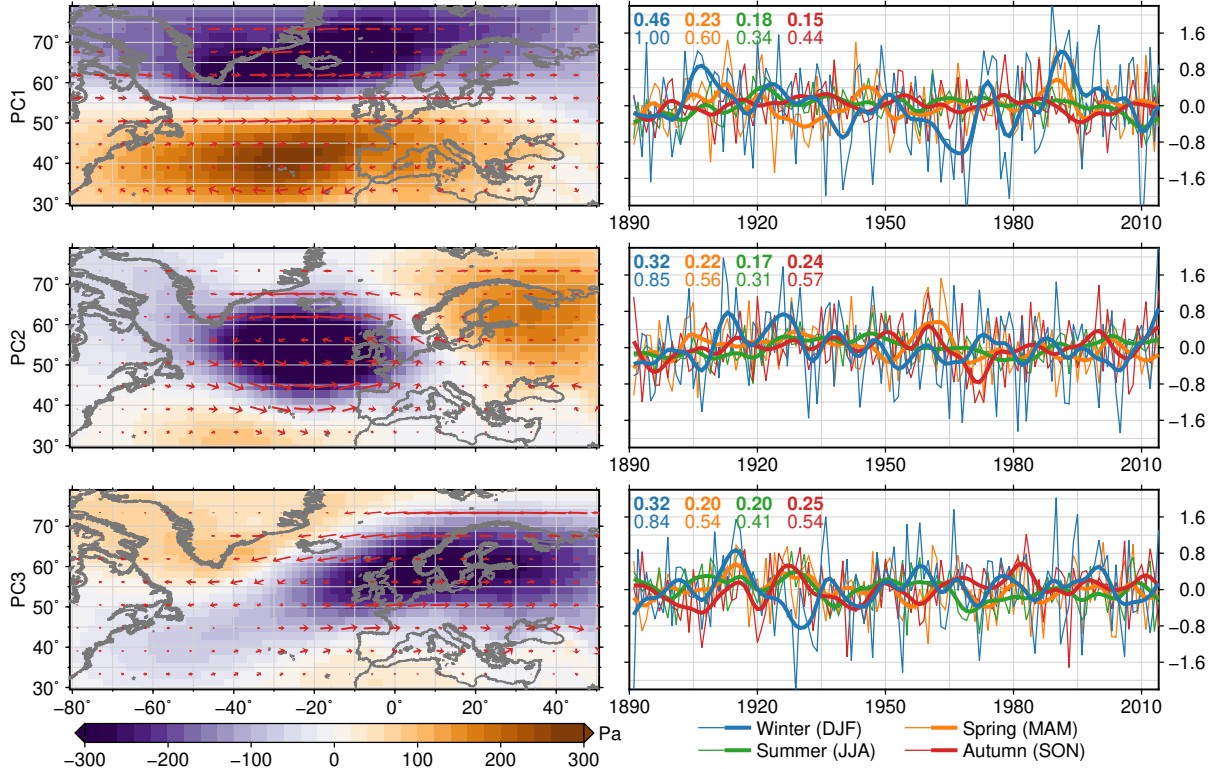

**Figure 5.** The first three Empirical Orthogonal Functions (EOFs) and accompanying Principal Components (PCs) of sea-level pressure above the North Atlantic Ocean. The left panels show the spatial patterns of each EOF. The red lines depict the associated geostrophic wind vectors. The right panels show the season-mean PCs (thin line), and the season-mean PCs after applying a 10-year low pass filter. The variance of the monthly-mean principal components is scaled to 1, and the numbers in the top left corner denote the low-pass filtered variance (bold) and the unfiltered variance (regular) for each season. Multiplication of the EOF with the accompanying PC gives the resulting sea-level pressure anomaly in Pascal.

Scandinavia Pattern respectively (e.g. Cassou et al., 2004). It must be noted that the principal components computed here, are not fully interchangeable with the commonly used original indices, which are generally computed using different methods. The first EOF is associated by westerly winds over the whole North Sea basin, while the second EOF shows a more meridionally-oriented wind effect, albeit with a curvature over the North Sea. EOF3 has its center of action over southern Scandinavia, and

5    hence, the wind strength associated with this EOF shows a large north-south gradient over the North Sea, with stronger winds in the south. All three patterns show both season-to season variability (thin lines in Figure 5), as well as variability on multi-decadal scales (thick lines). For all three principal components, the variability in winter is the largest, both for the seasonal and low-pass filtered time series. The difference is most pronounced for the first principal component, which is a well-known feature of the NAO (Hurrell et al., 2003), but it's visible in all three PC's at seasonal and the low-pass filtered time series.





The three PCs are used in the large-scale regression model (Equation 5), whose results are depicted in the bottom row of Figure 7. Like the local regression model, the large-scale regression model also explains a large fraction of the multi-decadal variability in winter and autumn sea level, while for spring and summer, the explained fraction is substantially smaller. For some stations in winter, the large-scale model even explains more variability than the local model, which is especially the case for the stations around the southern part of the North Sea. This difference may have its cause in the complex wind and pressure patterns generated by the large-scale atmospheric patterns, which may not be well-captured by the single-point wind- and pressure time series used in Equation 4. For some other stations, such as Brest and Newlyn, the local model explains more variability than the large-scale model, suggesting that not all variability is driven by large-scale patterns, but local effects also play a role.

The fact that both models explain a large fraction of the variability in autumn and winter shows that the variability is predominantly driven by wind- and pressure changes that are linked to large-scale atmospheric patterns. Interestingly, for the Southern part of the North Sea, the large-scale model explains a larger fraction of the variability than the local model.

Not only the amplitude of the variability (Figure 4, but also the temporal pattern differs between stations in this region, which can be seen in Figure 6. This figure shows time series of long-term winter-mean sea-level variability together with the results from both regression models at twelve representative stations. This figure again shows the major features of Figure 2 and 4: high variability in the German Bight, low variability along the British coast and the Southern North Sea and the ability of both regression models to explain a large part of this variability. The pattern of variability shows differences over the region: the stations in the German Bight (Delfzijl, Cuxhaven, Esbjerg) and Oslo show a coherent variability pattern, which differs from the patterns found in other locations. For example, during the period 1985-2005, most stations along the eastern North Sea coast show an above-average sea level, which is much less pronounced along the British coast, and even corresponds to a drop in seasonal sea level for the stations Brest and Newlyn. For all these stations, both regression models explain these features, which shows that the differences between these stations must be caused by a difference in wind and pressure forcing, and consequently, the large-scale atmospheric patterns affect different stations in a different way, and the variability at different stations may be attributable to different combination of influence from the large-scale patterns.

To understand these differences between the forcing mechanisms between the different stations, Figure 7 shows the fraction of explained variance for each individual regressor and the full regression model of low-pass filtered DJF sea level both the local and large-scale regression models. This figure shows that the origins of the forcing differ substantially throughout the region: while the stations from the German Bight towards the Southern North Sea are dominated by zonal wind stress, the more northern stations are forced by a combination of zonal and meriodional wind and sea-level pressure. At the other side of the English Channel, Brest and Newlyn are dominated by sea-level pressure effects. This is also the case for Torshavn, which can be explained by the fact that Torshavn is an off-shelf island, for which wind stress won't cause large storm surges due to the large ocean depth and the absence of a large ocean boundary. Along the northern Norwegian coast, which also shows a large variability signal, both zonal and meridional wind, as well as surface pressure variability explain a large fraction of the variability.





**Figure 6.** Time series of winter (DJF) sea level, the reconstructed sea level from the regression model, and the residual at selected tide-gauge stations. A 10-year low pass filter has been applied to all time series.



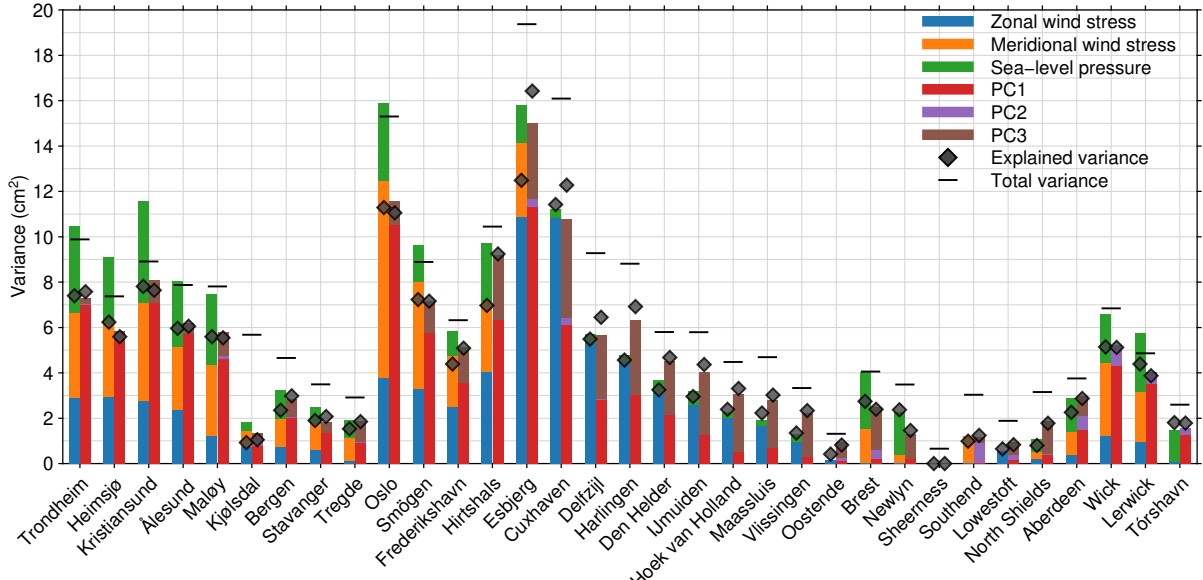

**Figure 7.** Total and explained variance of DJF sea level at each tide-gauge station. The left bar shows the variance explained by each term in the local model, and the right bar the variance explained by the large-scale model. Negative fractions of explained variance are not shown. The total and explained variances have been computed after applying a 10-year low-pass filter.

Despite the large regional variations in the local forcing agents, the first principal component PC1, which is closely tied to the North Atlantic Oscillation, explains the largest fraction of the variability for most stations. For the Southern North Sea, which are less affected by the westerlies associated with this PC (see Figure 5), the third PC explains a large part of the variability. The first PC is associated with both zonal, meridional and pressure changes along the Norwegian coast, which explains the

aforementioned impact of wind and pressure on the stations in that region. The third PC is associated with a strong zonal geostrophic wind component over the Southern North Sea. The second PC explains a small part of the variance, even though the signal does contain a considerable decadal winter-mean signal. The only exception are Brest and Newlyn, where the PC2 affects the zonal wind.

For the the autumn season, different factors affect the low-frequency variability, as shown in Figure 8. In autumn, the

variability, which is generally much smaller than in winter is generally driven by the same drivers as the winter variability, but the local wind and pressure variability is driven by a different combination of large-scale patterns for some stations, especially along the German Bight, where the seasonal variability is now mostly driven by the third PC instead of the first PC.

The trends in seasonal sea-level deviations, depicted in Figure 3, are also to a large extent caused by the atmospheric forcing. Figure 9 shows the same trends in seasonal deviation, but after removing sea-level deviations explained by the local and large-

scale model. The reduction in seasonal trends is largest in winter and autumn, and the seasonal trends have been reduced to the same order as typical secular mean sea level trends. Hence, although the simple regression model explains a large fraction of the variability, the unexplained variability still causes substantial seasonal trends.




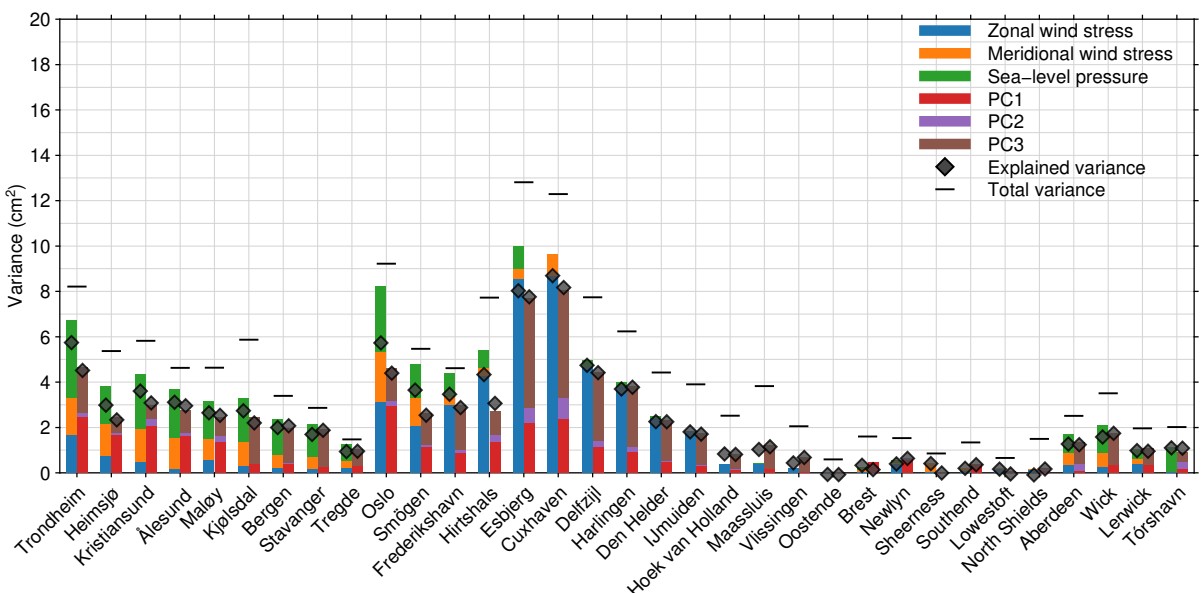

**Figure 8.** Same as Figure 7, but for the autumn (SON) season.

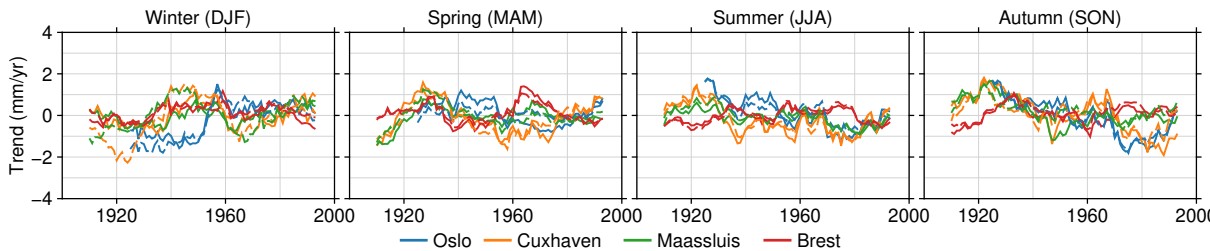

**Figure 9.** Running-mean 40-year trends in seasonal sea level deviations for four stations after removal of the local (dashed lines) and large-scale (solid lines) models. The top panels show trends in the original time series, and the bottom row shows the trends Trends are only shown for time windows with at least 30 years of data. Note that these trends have been computed from the time series without the low-pass filter applied.



# 4 Conclusions

In this paper, we have analysed the low-frequency variability in the seasonal deviations from annual mean sea level in the North Sea region. Low-frequency variability of winter-mean and autumn-mean sea level shows a spatially-varying pattern, with the highest values encountered along the German Bight. The major driving mechanism of this variability is wind and pressure. The wind generally plays a large role for locations that show large low-frequency variability, and is indeed weaker where wind plays a minor role, e.g. the British North Sea coast. The low-frequency changes in local wind and pressure are linked to large-scale atmospheric patterns, which resemble the NAO, EAP and Scandinavia patterns. Hence, the low-frequency variability in large-scale atmospheric patterns translate into low-frequency winter-mean and autumn-mean sea-level variability. In spring and summer, the low-frequency variability is smaller and can only to a small extent be explained by local and large-scale atmospheric forcing.

This seasonal sea-level variability is mostly caused by wind and pressure changes. Therefore, extreme sea levels associated with storm surge events are not superimposed onto this variability. In other words: a storm that occurs during a 'low phase' will not generate a lower surge level than when the same storm occurs during a 'high phase'. Because the sea-level response to local wind and pressure changes in the North Sea is mostly barotropic in nature (e.g. Chen et al., 2014; Dangendorf et al., 2014a), the typical sea-level adjustment time scale to wind and pressure changes will not be longer than a few days (Dimon et al., 1997). However, the sea-level response to atmospheric forcing is not strictly barotropic in the North Sea (Tsimplis et al., 2006; Calafat et al., 2012), and local and large-scale atmospheric changes do not explain all variability. As such, seasonal sea-level changes could still play a role in variability in storm-surge heights.

Multi-decadal seasonal sea-level variability in the North Sea is of the same order of magnitude as the long-term trend in mean sea level, and as a result, multi-decadal trends in annual mean sea level are in general not representative for the trends of the winter record in isolation. For processes that rely on long-term sea level variability, for example coastal sand suppletion this difference needs to be taken into account. Since a large part of the variability can be explained by a simple regression model, this correction should not pose a challenge.

*Code and data availability.* No substantial data sets and computer codes have been generated for this study.

*Author contributions.* T.G. conceived the idea for the study. T.F. did the analysis and made the figures. Both authors contributed to the analysis of the results and wrote the article.

*Competing interests.* The authors declare no competing interest





*Acknowledgements.* The tide-gauge data has been obtained from the Permanent Service for Mean Sea Level (PSMSL, www.psmsl.org). The NOAA 20th Century Reanalysis V2 data has been downloaded from www.esrl.noaa.gov/psd/data/20thC_Rean/. The principal component analysis has been conducted using eofs version 1.3 (Dawson, 2016). All figures have been produced with the Generic Mapping Tools (Wessel et al., 2013). Part of this research (T.F.) was carried out at the Jet Propulsion Laboratory, California Institute of Technology, under a contract
5  with the National Aeronautics and Space Administration.



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
