# Peer review of "Multidecadal variability in seasonal mean sea level along the North Sea coast"

_Ocean Science, 2018_

## Referee Comment (RC1) · Anonymous Referee #1 · 26 Sep 2018

The manuscripts is focused on the amplitude of decadal and multidecadal variability of sea-level in the North Sea and its connections to the atmospheric forcing. The background motivation is to set a backdrop to frame the observed long-term trends of sea-level rise. The main conclusion is that wind forcing and sea-level-pressure variability are important contributors to the decadal sea-level variability, and that this variability is larger in the winter season. Methodologically, the study tries to separate the local atmospheric forcing from the impact of large-scale atmospheric patterns, and concludes that the relative importance of these two factors varies regionally. Also, the relative impact of the large-scale atmospheric patterns can be different depending on the location of the tide-gauge.

My overall impression of the manuscript is positive. It is in general clearly written and

well structured. The motivation and conclusions should be interesting also for a wider readership. I have only some minor comments that the authors may want to consider in a slightly revised version.

1. My most important and more general comments pertains the separation of the atmospheric drivers between local and large-scale atmospheric patterns. I guess that both are not totally independent, at least the statistical models used doe not explicitly attempt to separate the influence of both. Clearly, the large-scale patterns, such as the NAO, are also related to local changes in sea-level-pressure and local wind-stress, so that the amount of variances explained by local and large-scale patterns partially overlap . Could this overlap be quantified, or could these factors be statistically separated in a statistical model that uses all predictors simultaneously ?

2. Page 2, line 10 The the sea-level response to atmospheric pressure often deviates from the inverse barometer effect (Woodworth, 2017b). I guess the authors mean a purely equilibrium inverse barometer effect, and that this deviation is caused by the time scales considered here (seasonal means) that indicate that the response of sea-level to changes in atmospheric pressure do not reach equilibrium. Is this the cause or are there other causes ? The authors may want to help the reader here.

3. Page 10 line 5 There is a duplication in this paragraph ' generally only explain a small part of the variability explain'

4. Page 8, line 1 It must be noted that the principal components computed here, are not fully interchangeable ... Delete comma after 'here'

5. Figure3. The plots show the gliding linear trends. Actually, these plots do not only show the variability of the trends but also the long-term acceleration (or lack thereof). According to these plots, only the winter records of Oslo and Cuxhaven show an acceleration of the sea-level rate, whereas all for other records the rate is statistically flat. Is this interpretation correct ? If yes, it may be worth a short comment. Also related to this, it may be worth to include in the plot error bars in the estimation of the trends.

These error bars would be dependent on the period and record, but to include a shading around the main lines could clutter the plot. My suggestion would be to include a typical error-bar as side vertical segment just as a guideline for the reader.

6. Figure 10 Related to my previous comment, it seems that after removing the impact of atmospheric forcing, only the Oslo winter record show a long-term acceleration. Would this be statistically significant ? would it be worth mentioning ?

---

## Referee Comment (RC2) · Anonymous Referee #2 · 27 Sep 2018

This study investigates the importance of interannual variability in seasonal sea level records in the North Sea region and it's relation to atmospheric drivers. To do so, long-term tide gauge records are analysed and compared to atmospheric forcing agents such as local wind stress and sea level pressure as well as large scale atmospheric patterns. The authors find that variability is largest for the winter season and show, by using a simple linear regression model, that a large part of this variability can be explained by atmospheric forcing – though varying regionally. Other possible drivers are not discussed. The study expands on previous studies that were confined to one record or focused on annual means of sea level.

In that respect, it is an interesting study that deserves publication. I have a few, mostly minor, comments though that I think should be addressed. Most importantly, the de-

scription of the analysis could be improved to be more concise and consistent. In particular, the relation between local wind and pressure should be examined in more detail, as they are both used as predictors in the model and are surely not independent of each other. For this, and other comments, see below.

(Page and line numbers refer to the manuscript as downloaded from https://www.ocean-sci-discuss.net/os-2018-102/os-2018-102.pdf)

Line 2: "... we quantify low-frequency seasonal variations from annual-mean sea level..." - I'm not sure I understand what you mean. Maybe "quantify the relative importance of seasonal variations to annual variability" or "separate seasonal variability from annual variability"?

Line 8: "... which have their cause in ..." - although it makes sense, I would be careful with implying causal relationships from the analysis in this study. We are actually talking about two interpretations of the same thing. I would suggest you replace the phrase with "which are related to". Line 20: Replace "with a view" with "with respect"

Line 17: Do you mean the NAO or really the North Atlantic Ocean?

Line 24: maybe clarify a bit? What do you mean by "The difference of the (...) difference"?

Line 3: Is the year missing after PSMSL? Also, I can't find PSMSL in the References. You can check the psmsl referencing recommendation here: http://psmsl.org/data/obtaining/reference.php

Line 6: "...a single record..." without the "s"

Line 6: "... over the period..." or shorten to "over the common period".

Line 26: "average to get seasonal anomalies": the same way you treated the tide gauge data, i.e. you subtract the annual mean and then average over the seasons?

Equation 4: I can imagine that sea-level pressure and wind stress are correlated? Did you test for this? In that case, did you do a stepwise regression in order to check whether the inclusion of additional predictors increases the explained variance significantly?

Line 9-14: It is implied that you use monthly data, but please mention it for completeness. Also, it might not be clear for the reader why you do not treat the reanalysis data the same way as the other data sources. For example, why do you remove the seasonal cycle, do you detrend it before the EOF analysis, and which data do you use in Equation 5: surely, you have to compute season-mean PCs in order to use the regression model?

Line 29: Replace while with with.

Figure 2: I would prefer to have the label standard deviation next to the colorbar.

Line 3: "... each season..." without the "s"

Line 5: Here, I'm not sure but I would suggest to replace between with in.

Line 12: Maybe add "after vertical land movement due to GIA has been accounted for"? Or "secular trend in absolute sea level".

Line 6: "In spring and summer, ..." - there is something wrong with this sentence. Please check.

Figure 5, caption: Instead of "red lines" you might as well write "red arrows".

Line 3: Replace by with with.

Line 1: I would suggest to replace "whose results" with "and the results"

Line 11: "Interestingly,..." This sounds as if you mentioned the fact for the first time, but in fact you have done so already in the previous paragraph (Line 5). I would rewrite this sentence together with the next ones. They seem, somehow not consistent: you mention that amplitude and temporal behaviour differ in this region (Line 13) but it is a bit unclear which region you are referring to. The southern part of the North Sea?

Line 13: A parenthesis is missing after Figure 4.

Line 25-26: How did you compute the fraction of explained variance of each predictor? Did you use the regression coefficients computed from Equation 4 and 5 (so, the full model), or did you perform the regression again with only one predictor? The results might be different, especially if the predictors (pressure and wind stress) are correlated.

Figure 6 (and other Figures): it would be easier for the reader who is not that familiar with the region if you could also include the station numbers, not just the name, in the figures. This way it would be easier to find the location of the station by looking only at the upper part of Figure 1.

Figure 7: If this is only about the relative importance of the different predictors, I would consider to normalize the explained variances with the total variance. That way, differences between stations are more obvious and easier to assess.

Line 3: Replace are (first word) with is.

Figure 9, caption: There are no top or bottom panels.

Line 3: "The major driving mechanism...." - at the risk of being pedantic, wind and pressure are not really a mechanism but rather a forcing. The question is through which mechanisms these forcing agents cause the variability in seasonal sea level.

Line 13-17: First you state that the sea-level response to local changes in wind and pressure is mostly barotropic, only to write in the next sentence that it isn't strictly barotropic. While there is no contradiction, the reader might be confused about what you are aiming at (I was). I would suggest to rephrase the sentence.

Line 17: "... and local and large scale atmospheric changes do not explain all variability." - Can you think of other processes that could be responsible for the variability in seasonal sea level?

Line 20: "... multi-decadal trends in annual sea level are in general not representative for the trends of the winter record..." - I agree to this general statement! But is it true for this region? It seems that the variability in winter could dominate the interannual variability as it is so large compared to the other seasons (except autumn). Have you looked at annual multi-decadal trends and compared them to the winter multi-decadal trends? Maybe, in this region, they are very similar and one might indeed infer winter decadal variability from annual means?

---

## Author Comment (AC1) · 7 Nov 2018

The manuscript is focused on the amplitude of decadal and multidecadal variability of sea-level in the North Sea and its connections to the atmospheric forcing. The background motivation is to set a backdrop to frame the observed long-term trends of sea-level rise. The main conclusion is that wind forcing and sea-level-pressure variability are important contributors to the decadal sea-level variability, and that this variability is larger in the winter season. Methodologically, the study tries to separate the local atmospheric forcing from the impact of large-scale atmospheric patterns, and concludes that the relative importance of these two factors varies regionally. Also, the relative impact of the large-scale atmospheric patterns can be different depending on the location of the tide-gauge.

My overall impression of the manuscript is positive. It is in general clearly written and well structured. The motivation and conclusions should be interesting also for a wider readership. I have only some minor comments that the authors may want to consider in a slightly revised version.

1. My most important and more general comments pertains the separation of the atmospheric drivers between local and large-scale atmospheric patterns. I guess that both are not totally independent, at least the statistical models used do not explicitly attempt to separate the influence of both. Clearly, the large-scale patterns, such as the NAO, are also related to local changes in sea-level-pressure and local wind-stress, so that the amount of variances explained by local and large-scale patterns partially overlap. Could this overlap be quantified, or could these factors be statistically separated in a statistical model that uses all predictors simultaneously?
The local and large-scale models are indeed not independent: the large-scale atmospheric patterns and local wind and pressure changes are linked. Therefore, the vast part of the local wind and pressure changes can be explained by the large-scale patterns. The variability that is explained by the local wind and pressure changes after removing the variability resulting from the large-scale atmospheric variability is negligibly small. We have added an extra remark to make explicitly clear that large-scale atmospheric patterns and local wind and pressure are related quantities:
P2L22: These indices characterize the prevailing large-scale patterns in the atmosphere, but on a regional level, they translate into regional wind- and sea-level pressure changes that induce local sea-level variations.

2. Page 2, line 10 The sea-level response to atmospheric pressure often deviates from the inverse barometer effect (Woodworth, 2017b). I guess the authors mean a purely equilibrium inverse barometer effect, and that this deviation is caused by the time scales considered here (seasonal means) that indicate that the response of sea level to changes in atmospheric pressure do not reach equilibrium. Is this the cause or are there other causes? The authors may want to help the reader here.
Yes, we indeed meant that around the North Sea, the local relation between sea level and local atmospheric pressure is often different from what would be expected from the static equilibrium expected from the inverted barometer effect. The most likely cause for this departure is that pressure changes co-vary with local and remote winds, precipitation etc., which also affect sea level via surges, for example (Woodworth 2009). Since the adjustment time scale for the IB effect is generally much shorter than a season (Wunsch and Stammer, 1997), it is unlikely that this departure is linked to the time scales we are interested in. We have added an explanation on the possible reasons for this departure:
P2L13: It is known that the sea-level response to local atmospheric pressure variability along the British coast often deviates from the equilibrium inverse barometer effect, which has its likely cause in factors that co-vary with atmospheric pressure, such as wind-induced surges (Woodworth et al., 2009; Woodworth, 2017b).

3. Page 10 line 5 There is a duplication in this paragraph 'generally only explain a small part of the variability explain'
Fixed

4. Page 8, line 1 It must be noted that the principal components computed here, are not fully interchangeable ... Delete comma after 'here'
Fixed

5. Figure3. The plots show the gliding linear trends. Actually, these plots do not only show the variability of the trends but also the long-term acceleration (or lack thereof). According to these plots, only the winter records of Oslo and Cuxhaven show an acceleration of the sea-level rate, whereas all for other records the rate is statistically flat. Is this interpretation correct? If yes, it may be worth a short comment. Also related to this, it may be worth to include in the plot error bars in the estimation of the trends. These error bars would be dependent on the period and record, but to include a shading around the main lines could clutter the plot. My suggestion would be to include a typical errorbar as side vertical segment just as a guideline for the reader.

Thank you very much for bringing up this comment and number 6. The plot indeed suggests accelerations in seasonal sea level for some stations, which to a large extent disappear when removing the atmospheric forcing. Note that we do not try to explain sea-level accelerations in this paper, but we fully agree that this point must be discussed. We have chosen not to quantify the effect on sea-level accelerations, but to note that the low-frequency variability could be interpreted as an acceleration, and removing the wind and pressure effects reduces this low-frequency variability, and thus the possible accelerations induced by this variability. We have added remarks about this acceleration in seasonal sea level, and the effects of removing the atmospheric forcing.

P7L3: This variability in seasonal deviations could also be interpreted as an acceleration: for example, the trend in winter sea-level deviations in Oslo and Cuxhaven is generally higher in the last few decades than in the first few decades, which could translate into a long-term positive acceleration in the seasonal sea-level deviation time series.

P12L17: For all seasons, the local and large-scale regression model explain a large fraction the running-mean trends. For winter and autumn, after applying the regression models, the residual seasonal trends have the same order of magnitude as typical secular mean sea level trends, and the trend differences between the beginning and end of the considered periods have been substantially reduced. However, not all low-frequency has been explained by the models, and the residual time series still contain trends and accelerations.

Furthermore, to avoid that the trends in Figure 3 are mis-interpreted as long-term trends and accelerations in annual sea level, we added an extra remark:

P7L6: Again, note that these trends are trends in seasonal deviations from annual-mean sea level, and they don't represent secular trends in sea level.

About the uncertainty estimates: we have added a shading to Figures 3 and 9, which now shows confidence intervals from the estimated trends under the assumption that the residual can be described by an AR1 process. For Figure 9, to prevent clutter, we have split up the Figure into an upper and lower row. We have added a short remark on how these trends are computed to the Data and Methods section:

P6L3: To estimate trends in seasonal sea-level deviations, we use the Hector software (Bos et al., 2013), which computes the trend and the 1-sigma confidence intervals under the assumption that the residuals can be described by a first-order autoregressive (AR1) process.

We added a short sentence to discuss the reduction of the confidence intervals that results from applying the regression model:

P12L17: For winter and autumn, after applying the regression models, the residual seasonal trends have the same order of magnitude as typical secular mean sea level trends, and the

6. Figure 10 Related to my previous comment, it seems that after removing the impact of atmospheric forcing, only the Oslo winter record show a long-term acceleration. Would this be statistically significant ? would it be worth mentioning ?
Please see our remarks at point #5, which also handles this point.

---

## Author Comment (AC2) · 7 Nov 2018

This study investigates the importance of interannual variability in seasonal sea level records in the North Sea region and it's relation to atmospheric drivers. To do so, long-term tide gauge records are analysed and compared to atmospheric forcing agents such as local wind stress and sea level pressure as well as large scale atmospheric patterns. The authors find that variability is largest for the winter season and show, by using a simple linear regression model, that a large part of this variability can be explained by atmospheric forcing – though varying regionally. Other possible drivers are not discussed. The study expands on previous studies that were confined to one record or focused on annual means of sea level. In that respect, it is an interesting study that deserves publication. I have a few, mostly minor, comments though that I think should be addressed.

Most importantly, the description of the analysis could be improved to be more concise and consistent. In particular, the relation between local wind and pressure should be examined in more detail, as they are both used as predictors in the model and are surely not independent of each other. For this, and other comments, see below. (Page and line numbers refer to the manuscript as downloaded from https://www.ocean-sci-discuss.net/os-2018-102/os-2018-102.pdf)

**Page 1**

Line 2: "… we quantify low-frequency seasonal variations from annual-mean sea level…" - I'm not sure I understand what you mean. Maybe "quantify the relative importance of seasonal variations to annual variability" or "separate seasonal variability from annual variability"?
We agree that this sentence is indeed ambiguous. We have changed this sentence to:
P1L2: In this study, we quantify low-frequency variability in seasonal deviations from annual-mean sea level and look for drivers of this variability.

Line 8: "…which have their cause in …" - although it makes sense, I would be careful with implying causal relationships from the analysis in this study. We are actually talking about two interpretations of the same thing. I would suggest you replace the phrase with "which are related to".
We have changed the sentence:
P1L6: We find that the variability is much stronger in winter and autumn than in other seasons, and that this winter and autumn variability is predominantly driven by wind and sea-level pressure anomalies which are related to large-scale atmospheric patterns.

Line 20: Replace "with a view" with "with respect"
Fixed

**Page 2**

Line 17: Do you mean the NAO or really the North Atlantic Ocean?
Fixed

Line 24: maybe clarify a bit? What do you mean by "The difference of the (…) difference"?
We have re-phrased this sentence:
P2L25: Seasonal deviations from annual-mean sea level contain strong multi-decadal components, who manifest themselves as possible trends and accelerations in records that span multiple decades

**Page 3**

Line 3: Is the year missing after PSMSL? Also, I can't find PSMSL in the References. You can check the psmsl referencing recommendation here: http://psmsl.org/data/obtaining/reference.php
We have added the reference including extraction date to the bibliography

Line 6: "…a single record…" without the "s"
Fixed

Line 6: "… over the period…" or shorten to "over the common period".

Fixed

Line 26: "average to get seasonal anomalies": the same way you treated the tide gauge data, i.e. you subtract the annual mean and then average over the seasons?
Yes, this sentence has been altered to make the procedure clear:
P3L27: To obtain seasonal anomalies, we follow the same procedure as with the tide-gauge data: we remove the annual-mean and group the monthly anomalies into season-average anomalies.

**Page 5**
Equation 4: I can imagine that sea-level pressure and wind stress are correlated? Did you test for this? In that case, did you do a stepwise regression in order to check whether the inclusion of additional predictors increases the explained variance significantly?
We did not check for mutual correlations between regressors. We have implemented a stepwise regression model that subsequently adds regressors to the model and using an adjusted $R^2$ metric to check whether each new regressor explains a significant amount of variance. We have added a description of the followed procedure in the Data and Methods section. When updating the code, we discovered a bug which caused the wrong number was subtracted from the annual mean values. We have fixed this bug, double-checked the code, and updated all results with the improved code.

Line 9-14: It is implied that you use monthly data, but please mention it for completeness. Also, it might not be clear for the reader why you do not treat the reanalysis data the same way as the other data sources. For example, why do you remove the seasonal cycle, do you detrend it before the EOF analysis, and which data do you use in Equation 5: surely, you have to compute season-mean PCs in order to use the regression model?
We agree that is paragraph is sometimes unclear, and we have re-written the explanation of the procedure we followed. We do remove the trend and the mean seasonal cycle before computing the EOF analysis, as not removing de mean seasonal cycle results in a dominant EOF mode related to the mean seasonal cycle. This removal sounds contradictory, but we are interested in low-frequency deviations from the mean, and not in the mean seasonal cycle itself. We now explicitly mention this apparent contradiction in line xx:
P5L24: Note that removing the semi-annual and annual cycle does remove the mean cycles themselves, but not the low-frequency variability around the mean cycles, which is the quantity we are interested in.

Line 29: Replace while with with.
Fixed

**Page 6**
Figure 2: I would prefer to have the label standard deviation next to the colorbar.
Label has been moved

Line 3: "… each season…" without the "s"
Fixed

Line 5: Here, I'm not sure but I would suggest to replace between with in.
Fixed

Line 12: Maybe add "after vertical land movement due to GIA has been accounted for"? Or "secular trend in absolute sea level".
Fixed, used the phrase secular trend in geocentric sea level.

**Page 7**
Line 6: "In spring and summer, …": there is something wrong with this sentence. Please check.

Fixed

**Page 8**

Figure 5, caption: Instead of "red lines" you might as well write "red arrows".
Fixed

Line 3: Replace by with with.
Fixed

**Page 9**

Line 1: I would suggest to replace "whose results" with "and the results"
Fixed, also fixed the reference to the wrong figure.

Line 11: "Interestingly," This sounds as if you mentioned the fact for the first time, but in fact you have done so already in the previous paragraph (Line 5). I would rewrite this sentence together with the next ones. They seem, somehow not consistent: you mention that amplitude and temporal behaviour differ in this region (Line 13) but it is a bit unclear which region you are referring to. The southern part of the North Sea?
We have revised this section to remove these ambiguities:
P8L13: Interestingly, for some stations in winter, especially for the stations in Belgium and in the southern part of the Netherlands, the large-scale model even explains more variability than the local model. This difference may have its cause in the complex wind and pressure patterns generated by the large-scale atmospheric patterns, which may not be well-captured by the single-point wind- and pressure time series used in Equation 4. For some other stations, such as Brest and Newlyn, the local model explains more variability than the large-scale model, suggesting that not all variability is driven by large-scale patterns, but local effects also play a role. The fact that both models explain a large fraction of the variability in autumn and winter shows that the variability is predominantly driven by wind- and pressure changes that are linked to large-scale atmospheric patterns.

Line 13: A parenthesis is missing after Figure 4.
Fixed

Line 25-26: How did you compute the fraction of explained variance of each predictor? Did you use the regression coefficients computed from Equation 4 and 5 (so, the full model), or did you perform the regression again with only one predictor? The results might be different, especially if the predictors (pressure and wind stress) are correlated.
We computed the fraction of explained variance using the regression coefficients computed from the full model, with the rejected regressors (non-significant/co-varying) not considered. We have added an extra line in the figure caption to clarify this:
Fig. 7 caption: The explained variance is computed using the regression coefficients from the model with all accepted terms, and not by only regressing each individual term.

**Page 10**

Figure 6 (and other Figures): it would be easier for the reader who is not that familiar with the region if you could also include the station numbers, not just the name, in the figures. This way it would be easier to find the location of the station by looking only at the upper part of Figure 1.
We have added the station numbers to figures 3,6,7,8, and 9.

**Page 11**

Figure 7: If this is only about the relative importance of the different predictors, I would consider to normalize the explained variances with the total variance. That way, differences between stations are more obvious and easier to assess.

We have considered that option, but in the current setting, the plot also depicts the ratio of the total variances and explained variances between stations, which would not be visible if the variances would be normalized. Hence, we have decided to keep the current layout.

Line 3: Replace are (first word) with is.
Fixed

**Page 12**

Figure 9, caption: There are no top or bottom panels.
This plot has changed as a result from remarks of reviewer #1, so the figure now actually has top and bottom panels.

**Page 13**

Line 3: "The major driving mechanism...." - at the risk of being pedantic, wind and pressure are not really a mechanism but rather a forcing. The question is through which mechanisms these forcing agents cause the variability in seasonal sea level.
We have re-phrased this sentence:
P13L6: This variability is largely forced by wind and pressure.

Line 13-17: First you state that the sea-level response to local changes in wind and pressure is mostly barotropic, only to write in the next sentence that it isn't strictly barotropic. While there is no contradiction, the reader might be confused about what you are aiming at (I was). I would suggest to rephrase the sentence.
We have re-phrased the sentence:
P14L2: However, the sea-level response to atmospheric forcing is mostly, but not necessarily fully barotropic in the North Sea

Line 17: "... and local and large scale atmospheric changes do not explain all variability." - Can you think of other processes that could be responsible for the variability in seasonal sea level?
We have added two examples of different processes that could play an additional role:
P14L4: For example, processes like ocean circulation changes and low-frequency variability in freshwater fluxes from rivers and locks (e.g. Gerkema and Duran-Matute, 2017) could drive low-frequency sea-level deviations. As such, seasonal sea-level changes could still play a role in variability in storm-surge heights.

Line 20: "... multi-decadal trends in annual sea level are in general not representative for the trends of the winter record..." I agree to this general statement! But is it true for this region? It seems that the variability in winter could dominate the interannual variability as it is so large compared to the other seasons (except autumn). Have you looked at annual multi-decadal trends and compared them to the winter multi-decadal trends? Maybe, in this region, they are very similar and one might indeed infer winter decadal variability from annual means?
We haven't looked into the representativeness of winter-mean sea level for annual sea level, the latter indeed being likely affected by winter-mean sea level (hence the caveat of our methodology denoted on P3L21). Although this relationship is likely there, since this paper deals with seasonal sea-level deviations, and not with annual-mean sea-level variability, we have decided not to quantify the impact of individual seasons on annual-mean sea level. We have added an extra sentence to the conclusions that seasonal variability will eventually affect annual-mean sea level:

[revised manuscript text omitted]